# 2-Hydroxyoleic Acid as a Self-Assembly Inducer for Anti-Cancer Drug-Centered Nanoparticles

**DOI:** 10.3390/ph16050722

**Published:** 2023-05-09

**Authors:** Antonia I. Antoniou, Giulia Nordio, Maria Luisa Di Paolo, Eleonora Colombo, Beatrice Gaffuri, Laura Polito, Arianna Amenta, Pierfausto Seneci, Lisa Dalla Via, Dario Perdicchia, Daniele Passarella

**Affiliations:** 1Dipartimento di Chimica, Università degli Studi di Milano, Via Golgi 19, 20133 Milano, Italy; antonia.antoniou@unimi.it (A.I.A.);; 2Dipartimento di Scienze del Farmaco, Università degli Studi di Padova, Via F. Marzolo 5, 35131 Padova, Italy; 3Dipartimento di Medicina Molecolare, Università degli Studi di Padova, Via G. Colombo 3, 35131 Padova, Italy; 4Istituto di Scienze e Tecnologie Chimiche “Giulio Natta”, SCITEC-CNR, Via G. Fantoli 16/15, 20138 Milano, Italy

**Keywords:** 2-hydroxyoleic acid (2OHOA), methyl 2-hydroxyoleate, nanoassemblies, conjugates, anticancer drugs, biphasic mesothelioma (MSTO-211H), colorectal adenocarcinoma (HT-29), glioblastoma (LN-229)

## Abstract

A potent nontoxic antitumor drug, 2-hydroxyoleic acid (**6**, 2OHOA) used for membrane lipid therapy, was selected as a self-assembly inducer due to its ability to form nanoparticles (NPs) in water. For this purpose, it was conjugated with a series of anticancer drugs through a disulfide-containing linker to enhance cell penetration and to secure drug release inside the cell. The antiproliferative evaluation of the synthesized NP formulations against three human tumor cell lines (biphasic mesothelioma MSTO-211H, colorectal adenocarcinoma HT-29, and glioblastoma LN-229) showed that nanoassemblies **16**–**22a,bNPs** exhibit antiproliferative activity at micromolar and submicromolar concentrations. Furthermore, the ability of the disulfide-containing linker to promote cellular effects was confirmed for most nanoformulations. Finally, **17bNP** induced intracellular ROS increase in glioblastoma LN-229 cells similarly to free drug **8**, and such elevated production was decreased by pretreatment with the antioxidant *N*-acetylcysteine. Also, nanoformulations **18bNP** and **21bNP** confirmed the mechanism of action of the free drugs.

## 1. Introduction

Drug delivery using nanomaterials has shown promising potential in creating more efficient systems helping to target drugs to specific tissues and cells [1,2,3]. For several years, we have been interested in the synthesis of natural products-based nanoparticles (NPs) for selective targeting and controlled release of anticancer drugs to tumors, limiting their toxicity against normal cells. More specifically, they were designed and synthesized conjugates able to form NPs that release drugs in cellular media, hetero-NPs bearing two different drugs, fluorescent NPs obtained by mixing drug- and fluorophore-based conjugates, as well as folic acid based hetero-NPs to exploit active drug targeting [4,5,6,7,8]. Naturally occurring anticancer drugs with a well-documented biological activity against specific cancers were selected; the drug was conjugated through a linker to a lipophilic self-assembly inducer, with the aim to obtain NPs in water [9,10]. The proper choice of a linker and a self-assembly inducer is crucial in the engineering of the final NPs in terms of particle size, structural, chemical, mechanical, and biological properties. Moreover, an intrinsic biological activity against the same target and potential synergy with the carried drug could be a further benefit. In our previous works, the self-assembly inducing moiety was either squalene (**1**), 4-(1, 2-diphenylbut-1-en-1-yl)aniline (**2**), 20-hydroxyecdisone (**3**), or betulinic acid (**4**) [4,5,11,12]. In continuing our exploration of new self-assembly inducers, our attention was directed to the nontoxic lipophilic anticancer drug 2OHOA (**6**) [13], the α-hydroxy derivative of oleic acid (**5**) (Figure 1).

2OHOA showed promising anti-cancer effects in lung adenocarcinoma, leukemic, and glioma cancer cells, with several trials currently in progress [13,14,15]. Furthermore, it represents a conceptually new approach for treating cancer, called membrane lipid therapy (MLT) [16,17]; by its interaction with membrane lipids, it causes profound membrane lipid remodeling in cancer cells but not in normal cells, altering the organization of membrane micro-domains and finally impairing proliferation by arresting cell cycle progress from the G1 to S phase [13,18]. It also modulates the activity of sphingomyelin synthase (SMS1), increasing the concentration of sphingomyelin in cancer cells with high specificity and potency [19]. Moreover, cellular and molecular evidence have shown that 2OHOA induces autophagy in glioma cancer cells, constituting a novel therapeutic strategy to combat glioma in apoptosis-resistant tumor cells [20]. Another study revealed that (*S*) 2OHOA was responsible for an increased concentration of sphingomyelin; nevertheless, racemic 2OHOA [21] exhibited an almost identical IC_50_ in the inhibition of the growth of the lung adenocarcinoma cell line A549, consistent with the combined effect on membrane organization by enantiopure or racemic 2OHOA, and on sphingomyelin concentration by the *S* isomer. Furthermore, 2OHOA can depolarize mitochondrial inner membranes by acting as a cancer-selective mitochondrial oxidative phosphorylation (OxPhos) uncoupler, compromising glycolytic stress response, and enhancing endoplasmic reticulum (ER) stress [22].

Examples in the literature of the application of 2OHOA in the NP field are rather limited. It can be effectively incorporated in cationic liposomes that resulted to be quite specific in targeting the tumor vasculature [23,24]. Moreover, it can form pH-dependent nano-self-assemblies with glycerol mono-oleate, with the potential to selectively target the acidic extracellular pH environment of cancer tissues [25].

In connection to our previous studies and considering the current state-of-the-art literature, we conjugated 2OHOA through a disulfide-based linker which secures the drug release (Figure 2) with well-known anticancer drugs (**7**–**13**, Figure 3), and evaluated (a) their ability to self-assemble in water and (b) the antiproliferative activity of the obtained NPs against human tumor cell lines. The presence of a disulfide bond is crucial for triggered drug release at the tumor site as the disulfide bond is stable at physiological body temperature, pH, and oxidation environment, whereas it can be degraded by reducing agents such as glutathione (GSH). In particular, the intracellular concentration of GSH is much higher than the extracellular one, due to the high amount of GSH produced by the cancer cells in comparison to normal cells [26,27,28]. Thus, the disulfide-based crosslinking approach was considered the most suitable for the preparation of our conjugates.

## 2. Results and Discussion

The present study reports the preparation and biological evaluation of a novel class of methyl 2-hydroxy oleate self-assembly drug conjugates against three human cancer cell lines (biphasic mesothelioma MSTO-211H, colorectal adenocarcinoma HT-29, and glioblastoma LN-229). The hydrophobic tail of methyl 2-hydroxyoleate (**14**) was used as a building block that could secure self-assembly in water, whereas the disulfide-containing linker secures intracellular drug release. Seven anticancer drugs **7**–**13** (Figure 3) were used for the preparation of a small library of methyl oleate conjugates to be assembled.

### 2.1. Synthesis of Drug-Methyl Oleate Conjugates

The preparation of drug-2OHOA conjugates started with the synthesis of building block **14**. Thus, oleic acid was deprotonated to the corresponding dianion by the system lithium diisopropylamide/ *N*,*N*′-dimethylpropyleneurea (LDA/DMPU) and then was α-hydroxylated by reaction with oxygen, leading to 2OHOA **6**. Upon reaction of the latter with boric acid in methanol, the corresponding methyl ester **14** was selectively provided in a very good yield (Figure 1).

Subsequently, condensation of methyl ester **14** with an excess of either sebacic acid or 4,4′-dithiodibutyric acid using 1-ethyl-3-(3-dimethylaminopropyl)carbodiimide hydrochloride (EDC·HCl) in presence of a catalytic amount of 4-dimethylaminopyridine (DMAP) afforded, respectively, monoesters **15a** and **15b** in good yields. These two key intermediates were then engaged in a second Steglich-type esterification with selected anticancer drugs, providing conjugates **16**–**21a,b** in moderate to excellent yields (Figure 4). As to paclitaxel, its analogs cabazitaxel and docetaxel, and epothilone A conjugates (**16**–**18a,b** and **20a,b,** respectively, Figure 4), the regioselectivity of the acylation has been assessed by our group in previous work [4] and was confirmed by the ^1^H NMR spectra of final conjugates.

For the preparation of amides **22a,b**, *N*-desacetyl thiocolchicine (**23**) was obtained after standard deacetylation of (–)-thiocolchicine **13** [29]. Then, condensation of amine **23** with either acid **15a** or **15b** led to final target amide conjugates **22a** and **22b** in good yields (Figure 2).

### 2.2. Preparation and Characterization of Nanoparticles

Once we obtained the above conjugates, the corresponding nanosuspensions were prepared in accordance with standard solvent evaporation protocols [30] and characterized in terms of their physical–chemical properties. More specifically, dynamic light scattering (DLS) and Z-potential measurements were carried out on each NP sample after 10 minutes of sonication, providing the results shown in Table 1.

DLS confirmed the formation of nanoassemblies in an aqueous medium. Namely, the low polydispersity index values (PI < 0.2) indicated that each methyl oleate-linker-drug conjugate **16**–**22a,b** was able to give monodisperse suspensions of **16**–**22a,bNPs**, with hydrodynamic diameters (HDs) in a 110–460 nm range. Even though some dimensions are close to the higher end of NPs’ definition (500 nm), we expect them to be able to exert their action and be internalized in cells. Additionally, the Z-potential was negative (<−25 mV) for all nanoassemblies, suggesting that electrostatic repulsion contributes to their colloidal stability. In order to get more information on their morphology, transmission electron microscopy (TEM) analysis was performed on the two most active NPs. As indicated in Figure 5, nanoassemblies exhibited a spherical morphology; however, 17bNP seemed to be quite unstable under the electron beam as is evident from the TEM micrograph.

### 2.3. Biological Investigation

Nanoformulations and the corresponding free drugs were assayed for their antiproliferative effect on human MSTO-211H (biphasic mesothelioma), HT-29 (colorectal adenocarcinoma), and LN-229 (glioblastoma) human tumor cells. The obtained GI_50_ values, that is, the concentration of tested free drug/NP that induces a 50% decrease in cell number with respect to the control culture, are shown in Table 2, and cytotoxicity curves are reported in Figure 6 and Appendix A.

The seven drugs **7**–**13** confirmed their cytotoxicity towards tumor cells, as GI_50_ in the low micromolar or nanomolar range were obtained in all cell lines taken into consideration, also in accordance with our earlier data [31]. In detail, podophyllotoxin (**12**) and thiocolchicine (**13**) exerted the lowest antiproliferative effect with GI_50_ values ranging from 0.011 to 0.030 μM, whereas cabazitaxel (**8**) and docetaxel (**9**) appear as the most effective drugs, showing GI_50_ values from 0.7 to 2.0 nM. We also evaluated 2OHOA, and it showed GI_50_ >100 μM in all cell lines, in accordance with reported evidence [20].

Incubation of cells in the presence of **16**–**22a,bNPs** resulted, in all cases, in significant cytotoxicity, with GI_50_ values in the micromolar and submicromolar range. Interestingly, such cellular effect is more pronounced for nanoconjugates containing 4,4′-dithiodibutyric acid (**16b**–**22b**) with respect to those built with sebacic acid as linker (**16a**–**22a**). In fact, the noticeable contribution of the disulfide linker appears particularly significant for the **17a**,**bNP** and **18a**,**bNP** pairs, in which it induces a 6- to 25-fold increase in activity, as confirmed also by the cytotoxicity curves shown in Figure 6A and B, respectively. Otherwise, NPs containing the free drug linkers **15a,bNP**, are unable to induce cytotoxicity (Table 2).

For the most cytotoxic drugs **8** and **9**, and for the corresponding NPs built on the disulfide linker 1**7bNP** and **18bNP**, the antiproliferative effect was also tested on human nontumorigenic Met-5A (mesothelium) cells. The obtained results evidence, as expected, notable cytotoxicity induced by the drugs, with 0.0006 ± 0.0001 μM and 0.0032 ± 0.0013 μM GI_50_ values for **8** and **9**, respectively, and confirm the decreased cell effect in the presence of 1**7bNP** or **18bNP**, that is, GI_50_ 0.11 ± 0.05 μM and 0.17 ± 0.04 μM, respectively, in accordance with the results on human tumor cell lines (Table 2).

Based on these results and on the remarkable antiproliferative effect of **17bNP** on all cell lines, we investigated its mechanism of action with the aim to assess if the intracellular events induced by free cabazitaxel (**8**) are maintained by the nanoformulation **17bNP**. Taking into consideration that cabazitaxel promotes ROS production in human cancer cells, and this increase was correlated with its cytotoxic effect [32], LN229 cells were loaded with 2′,7′-dichlorofluorescein diacetate, treated with free **8** and **17bNP** for 30 min and then intracellular ROS production was determined by a fluorimetric assay. We found that both free drug and nanoformulation induced intracellular ROS accumulation in LN-229 glioblastoma cells, and such elevated ROS production was inhibited by pretreatment with the antioxidant N-acetylcysteine (NAC). These results confirm the ability of **17bNP** to cause a redox imbalance in cancer cells, in accordance with the mechanism of action of free drug **8** (Figure 7).

Intracellular effects were investigated also for **18bNP** and its free drug **9**. Based on the ability of docetaxel (**9**) to stabilize microtubules [33], we investigated cell cycle progression by flow cytometry (Figure 8A). The DNA content-based cell cycle analysis of LN229 cells after treatment with **9** revealed a decrease in the G0/G1 phase. 

And a greater proportion of cells in the mitotic phase (G2/M) with respect to the control, as expected. The incubation of cells with the conjugate1**8bNP** demonstrated a comparable behavior (Figure 8A). Similarly, podophyllotoxin (**12**), a microtubule-targeting agent that inhibits microtubule assembly [34], induced G2/M blockade in treated LN229 cells, as **21bNP**, thus confirming also for these conjugates the conservation of the mechanism of action of free drugs (Figure 8B).

## 3. Materials and Methods

### 3.1. General Methods

All reagents and dry solvents employed in the present work were commercially available and used without further purification. When required, reactions were carried out under a dry nitrogen atmosphere in pre-flamed glassware. Anhydrous Na_2_SO_4_ was used for drying solutions, and the solvents were then routinely removed at ca. 40 °C under reduced pressure using a rotary vacuum evaporator. Flash column chromatography (FCC) was performed on Merck silica gel 60 (240–400 mesh, Darmstadt, Germany), and analytical thin layer chromatography (TLC) was performed on Merck silica gel 60F254 (0.2 mm film, Darmstadt, Germany) pre-coated on aluminum foil. Spots on the TLC plates were visualized with UV light at 254 nm, and the TLC plate was stained with a solution of potassium permanganate.

^1^H NMR spectra were obtained at 400.15 and 300.14 MHz, and ^13^C NMR spectra at 100.63 and 75.47 MHz on a Bruker DRX-400 or DRX-300 spectrometer, respectively, in the indicated solvents. Chemical shifts (*δ*) are shown in parts per million (ppm) downfield from tetramethylsilane (TMS) and coupling constants (*J*) are reported in Hertz. Electrospray ionization (ESI) mass spectra were recorded on a Fisons MD800 spectrometer and electrospray ion trap on a Finnigan LCQ advantage Thermo-spectrometer, using MeOH as solvent.

### 3.2. Experimental Procedures

#### 3.2.1. Synthesis of 2-Hydroxyoleic Acid (**6**)

To a stirred solution of oleic acid (1.0 g, 3.54 mmol) in dry THF (10 mL), DMPU (0.47 mL, 3.89 mmol) and LDA 1 M in THF (8.4 mL, 8.4 mmol) were added dropwise, and the reaction mixture was heated at 50–55 °C for 30 min. Then, the solution was allowed to gradually cool down to room temperature and oxygen gas was bubbled into it for 30 min. Subsequently, the reaction was quenched with aqueous 3M HCl (10 mL), and THF was removed under reduced pressure. The mixture was extracted with DCM (3 × 20 mL). The combined organic phases were washed sequentially with aqueous 1 M NaHSO_3_ (30 mL) until pH 3, water (30 mL), and brine (30 mL), dried over Na_2_SO_4,_ and concentrated under reduced pressure. The residue thus obtained was purified by direct FCC (6:4→1:1 *n*-hexane/EtOAc) to afford upon solvent removal target compound **6** as a white solid (498 mg, 1.66 mmol, 45%). R_f_ (1:1:0.1 *n*-hexane/EtOAc/AcOH) = 0.2; ^1^H NMR (400 MHz, CDCl_3_) *δ* 5.39–5.29 (m, 2 H), 4.27 (dd, *J =* 7.6, 4.3 Hz, 1 H), 2.05–1.96 (m, 4 H), 1.90–1.80 (m, 1 H), 1.75–1.64 (m, 1 H), 1.52–1.39 (m, 2 H), 1.37–1.21 (m, 18 H), 0.88 (t, *J =* 6.7 Hz, 3 H); ^13^C NMR (101 MHz, CDCl_3_) *δ* 179.8, 130.0, 129.6, 70.3, 34.1, 31.9, 29.8, 29.7, 29.5, 29.3, 29.2, 29.1, 27.2, 27.1, 24.8, 22.7, 14.1; ESI-MS (*m/z*) [M+Na], [2M+Na] calculated for C_18_H_34_O_3_ 321.24, 619.49. Found 321.69, 619.59.

#### 3.2.2. Synthesis of Methyl (Z)-2-Hydroxyoctadec-9-Enoate (**14**)

To a solution of 2-hydroxyoleic acid (0.45 g, 1.51 mmol) in MeOH (6.5 mL), boric acid (93 mg, 1.51 mmol) was added at room temperature, and the reaction mixture was stirred for 48 h. After removal of the solvent under reduced pressure, the residue thus obtained was purified by direct FCC (6:4 *n*-hexane/EtOAc) to provide upon solvent removal target methyl ester **14** as a colorless oil (377 mg, 1.21 mmol, 80%). R_f_ (6:4 *n*-hexane/EtOAc) = 0.42; ^1^H NMR (400 MHz, CDCl_3_) *δ* 5.39–5.29 (m, 2 H), 4.18 (dd, *J =* 7.3, 4.2 Hz, 1 H), 3.78 (s, 3 H), 2.05–1.96 (m, 4 H), 1.83–1.72 (m, 1 H), 1.68–1.57 (m, 1 H), 1.35–1.21 (m, 2 H), 1.37–1.21 (m, 18 H), 0.88 (t, *J =* 7.1 Hz, 3 H); ^13^C NMR (101 MHz, CDCl_3_) *δ* 175.8, 130.0, 129.7, 70.4, 52.3, 34.4, 31.9, 29.7, 29.6, 29.5, 29.3, 29.2, 29.1, 27.2, 27.1, 24.7, 22.7, 14.1; ESI-MS (*m/z*) [M+Na] calculated for C_19_H_36_O_3_ 335.26. Found 335.57.

#### 3.2.3. General Procedure for Compounds **15a-b**

To a solution of ester **14** (440 mg, 1.41 mmol) in DCM (10 mL) a dicarboxylic acid (4.27 mmol), EDC·HCl (298 mg, 1.55 mmol), DMAP (17 mg, 0.141 mmol) and Et_3_N (1.59 mL, 11.39 mmol) were added and the reaction mixture was stirred at room temperature for 24 h. Aqueous 1 M HCl (15 mL) was added, and the mixture was extracted with DCM (3 × 20 mL). The combined organic phases were washed with brine (20 mL), dried over Na_2_SO_4,_ and concentrated under reduced pressure. The residue thus obtained was purified by direct FCC (8:2→7:3 *n*-hexane/EtOAc), obtaining either pure target compound **15a** or (7:3 *n*-hexane/EtOAc) pure target compound **15b**, both as colorless oils.

(E)-10-((1-methoxy-1-oxooctadec-9-en-2-yl)oxy)-10-oxodecanoic acid (**15a**): Yield = 64% (450 mg, 0.91 mmol); R_f_ (7:3 n-hexane/EtOAc) = 0.25; ^1^H NMR (400 MHz, CDCl_3_) *δ* 5.39–5.30 (m, 2 H), 4.98 (t, *J =* 6.6 Hz, 1 H), 3.73 (s, 3 H), 2.38 (td, *J =* 7.6, 3.8 Hz, 2 H), 2.34 (t, *J =* 7.6 Hz, 2 H), 2.05–1.96 (m, 4 H), 1.85–1.77 (m, 2 H), 1.62 (sextet, *J =* 7.4 Hz, 4 H), 1.37–1.22 (m, 29 H), 0.88 (t, *J =* 7.1 Hz, 3 H); ^13^C NMR (101 MHz, CDCl_3_) *δ* 180.0, 173.3, 170.9, 130.0, 129.6, 72.1, 52.1, 34.0, 33.9, 31.9, 31.1, 29.7, 29.6, 29.5, 29.3, 29.0, 29.0, 28.9, 28.9, 28.9, 27.2, 27.1, 25.1, 24.7, 24.6, 22.6, 14.1; ESI-MS (*m/z*) [M+Na] calculated for C_29_H_52_O_6_ 519.37. Found 519.60.

(Z)-4-((4-((1-methoxy-1-oxooctadec-9-en-2-yl)oxy)-4-oxobutyl)disulfaneyl)butanoic acid (**15b**): Yield = 68% (515 mg, 0.97 mmol); R_f_ (1:1 *n*-hexane/EtOAc) = 0.23; ^1^H NMR (400 MHz, CDCl_3_) *δ* 5.39–5.28 (m, 2 H), 4.99 (t, *J =* 6.6 Hz, 1 H), 3.74 (s, 3 H), 2.77–2.70 (m, 4 H), 2.59–2.46 (m, 4 H), 2.10–1.96 (m, 8 H), 1.86–1.78 (m, 2 H), 1.41–1.22 (m, 21 H), 0.88 (t, *J =* 7.1 Hz, 3 H); ^13^C NMR (101 MHz, CDCl_3_) *δ* 178.9, 172.5, 170.8, 130.0, 129.6, 72.3, 52.2, 37.6, 37.5, 32.3, 32.3, 31.9, 31.0, 29.7, 29.6, 29.5, 29.3, 29.0, 29.0, 27.2, 27.1, 25.1, 24.1, 23.8, 22.7, 14.5; ESI-MS (*m/z*) [M+Na] calculated for C_27_H_48_O_6_S_2_ 555.78. Found 555.76.

#### 3.2.4. General Procedure for Drug Conjugates **16–22a,b**

To a solution of **15a,b** (0.04 mmol) in DCM (1.0 mL), EDC·HCl (10.3 mg, 0.06 mmol), DMAP (3.7 mg, 0.03 mmol), and one of the selected drugs (0.04 mmol) were added, and the reaction mixture was stirred at room temperature for 1–46 h. Aqueous 1 M HCl (10 mL) was added, and the mixture was extracted with DCM (5 × 5 mL). The combined organic phases were dried over Na_2_SO_4_ and concentrated under reduced pressure. The residue thus obtained was purified by direct FCC to afford pure drug conjugates.

**16a**: White foam; Reaction time = 5 h; Yield = 72% (39.6 mg, 0.030 mmol); R_f_ (1:1 *n*-hexane/EtOAc) = 0.23; ^1^H NMR (400 MHz, CDCl_3_) δ 8.12 (d, *J* = 7.0 Hz, 2 H), 7.73 (d, *J* = 7.0 Hz, 2 H), 7.60 (tt, *J* = 1.4 Hz, 1 H), 7.53–7.46 (m, 3 H), 7.44–7.30 (m, 7 H), 6.90 (d, *J* = 9.0 Hz, 1 H), 6.29 (s, 1 H), 6.26 (t, *J* = 9.3 Hz, 1 H), 5.95 (dd, *J* = 9.2, 3.3 Hz, 1 H), 5.68 (d, *J* = 7.2 Hz, 1 H), 5.50 (d, *J* = 3.3 Hz, 1 H), 5.39–5.27 (m, 2 H), 4.98 (t, *J* = 6.6 Hz, 2 H), 4.44 (dd, *J* = 10.8, 6.6 Hz, 1 H), 4.31 (d, *J* = 8.3 Hz, 1 H), 4.20 (d, *J* = 8.3 Hz, 1 H), 3.81 (d, *J* = 7.1 Hz, 1 H), 3.72 (d, *J* = 1.3 Hz, 3 H), 2.61–2.48 (m, 2 H), 2.45 (s, 3 H), 2.41–2.34 (m, 4 H), 2.22 (s, 3 H), 2.19–2.11 (m, 1 H), 2.04–1.96 (m, 4 H), 1.94 (d, *J* = 1.5 Hz, 3 H), 1.91–1.77 (m, 4 H), 1.68 (s, 3 H), 1.62 (t, *J* = 7.4 Hz, 2 H), 1.57 (t, *J* = 7.1 Hz, 2 H), 1.39–1.24 (m, 29 H), 1.23 (s, 3 H), 1.13 (s, 3 H), 0.87 (t, *J* = 7.0 Hz, 3 H); ^13^C NMR (101 MHz, CDCl_3_) δ 203.9, 173.5, 172.8, 171.4, 171.1, 169.9, 168.2, 167.2, 167.1, 142.9, 137.1, 133.8, 132.9, 132.1, 130.3, 130.2, 129.7, 129.3, 129.1, 128.8, 128.6, 127.2, 126.6, 84.6, 81.2, 79.3, 76.6, 75.7, 75.2, 73.9, 72.2, 71.9, 58.6, 52.9, 52.3, 45.7, 43.3, 35.6, 34.0, 33.8, 32.0, 31.2, 29.9, 29.7, 29.6, 29.4, 29.4, 29.1, 29.1, 29.1, 29.0, 27.3, 27.2, 26.9, 25.2, 24.8, 24.8, 22.8, 22.8, 22.3, 20.9, 14.9, 14.2, 9.7; ESI-MS (*m*/*z*) [M+Na] calculated for C_76_H_101_NO_19_ 1354.69. Found 1354.38.

**16b**: White foam; Reaction time = 5 h; Yield = 99% (53 mg, 0.039 mmol); R_f_ (1:1 *n*-hexane/EtOAc) = 0.22; ^1^H NMR (400 MHz, CDCl_3_) *δ* 8.13 (d, *J =* 7.0 Hz, 2 H), 7.74 (d, *J =* 7.0 Hz, 2 H), 7.60 (tt, *J =* 1.5 Hz, 1 H), 7.54–7.47 (m, 3 H), 7.44–7.32 (m, 7 H), 6.93 (d, *J =* 9.3 Hz, 1 H), 6.30 (s, 1 H), 6.25 (t, *J =* 8.9 Hz, 1 H), 5.97 (dd, *J =* 9.1, 3.4 Hz, 1 H), 5.68 (d, *J =* 7.1 Hz, 1 H), 5.51 (d, *J =* 3.1 Hz, 1 H), 5.39–5.27 (m, 2 H), 5.00–4.93 (m, 2 H), 4.44 (dd, *J* = 10.9, 6.7 Hz, 1 H), 4.31 (d, *J =* 8.2 Hz, 1 H), 4.20 (d, *J =* 8.4 Hz, 1 H), 3.81 (d, *J =* 7.1 Hz, 1 H), 3.70 (d, *J =* 2.8 Hz, 3 H), 2.74–2.47 (m, 9 H), 2.46 (s, 3 H), 2.42–2.33 (m, 1 H), 2.22 (s, 3 H), 2.20–2.12 (m, 1 H), 2.06–1.96 (m, 8 H), 1.91–1.76 (m, 4 H), 1.68 (s, 3 H), 1.42–1.22 (m, 21 H), 1.23 (s, 3 H), 1.13 (s, 3 H), 0.87 (t, *J =* 6.7 Hz, 3 H); ^13^C NMR (101 MHz, CDCl_3_) *δ* 203.9, 172.7, 172.1, 171.3, 170.7, 169.9, 168.1, 167.2, 167.1, 142.9, 137.1, 133.8, 133.7, 132.9, 132.1, 130.3, 130.2, 129.7, 129.3, 129.2, 128.9, 128.6, 127.2, 126.6, 84.6, 81.2, 79.2, 76.5, 75.7, 75.2, 74.2, 72.5, 72.2, 71.9, 58.6, 52.8, 52.3, 45.7, 43.3, 37.5, 37.1, 35.7, 35.6, 32.4, 32.1, 31.1, 29.9, 29.7, 29.6, 29.4, 29.4, 29.1, 29.1, 27.3, 27.2, 26.9, 25.2, 24.2, 24.0, 22.8, 22.8, 22.3, 20.9, 14.9, 14.2, 9.71; ESI-MS (*m/z*) [M+Na] calculated for C_74_H_97_NO_19_S_2_ 1390.60. Found 1390.72.

**17a***:* White foam; Reaction time = 1 h; Yield = 89% (46.8 mg, 0.036 mmol); R_f_ (1:1 *n*-hexane/EtOAc) = 0.34; ^1^H NMR (400 MHz, CDCl_3_) *δ*
^1^H NMR (400 MHz, CDCl_3_) *δ* 8.10 (d, *J =* 7.1 Hz, 2 H), 7.59 (t, *J =* 7.3 Hz, 1 H), 7.49 (t, *J =* 8.0 Hz, 2 H), 7.38 (t, *J =* 7.4 Hz, 2 H), 7.30 (t, *J =* 8.0 Hz, 3 H), 6.25 (t, *J =* 9.3 Hz, 1 H), 5.64 (d, *J =* 7.0 Hz, 1 H), 5.45 (br s, 1 H), 5.39–5.27 (m, 4 H), 4.98 (t, *J =* 6.7 Hz, 2 H), 4.82 (s, 1 H), 4.30 (d, *J =* 8.3 Hz, 1 H), 4.16 (d, *J =* 8.4 Hz, 1 H), 3.89 (dd, *J* = 10.7, 6.4 Hz, 1 H), 3.84 (d, *J =* 7.1 Hz, 1 H), 3.72 (s, 3 H), 3.43 (s, 3 H), 3.29 (s, 3 H), 2.74–2.63 (m, 1 H), 2.43 (s, 3 H), 2.41–2.27 (m, 5 H), 1.99 (s, 6 H), 1.85–1.75 (m, 3 H), 1.71 (s, 3 H), 1.66–1.60 (m, 3 H), 1.56–1.50 (m, 2 H), 1.42–1.35 (m, 3 H), 1.34 (s, 9 H), 1.32–1.17 (m, 33 H), 0.87 (t, *J =* 7.0 Hz, 3 H); ^13^C NMR (101 MHz, CDCl_3_) *δ* 205.0, 173.4, 172.8, 171.0, 169.7, 168.4, 167.0, 155.2, 139.6, 135.0, 133.6, 130.2, 130.1, 129.6, 129.3, 128.8, 128.6, 128.2, 126.3, 84.2, 82.5, 81.6, 80.7, 80.4, 78.9, 76.5, 74.8, 74.2, 72.1, 72.0, 57.2, 57.1, 56.8, 52.2, 47.4, 43.3, 35.0, 33.9, 33.7, 32.0, 31.9, 31.1, 29.8, 29.6, 29.5, 29.3, 29.3, 29.0, 28.8, 28.2, 27.2, 27.1, 26.7, 25.1, 24.8, 24.6, 22.8, 22.7, 21.0, 14.1, 14.2, 10.4; ESI-MS (*m/z*) [M+Na] calculated for C_74_H_107_NO_19_ 1336.73. Found 1336.62.

**17b**: White solid; Reaction time = 4 h; Yield = 90% (48.6 mg, 0.036 mmol); R_f_ (1:1 *n*-hexane/EtOAc) = 0.3; ^1^H NMR (400 MHz, CDCl_3_) *δ*
^1^H NMR (400 MHz, CDCl_3_) *δ* 8.11 (d, *J =* 8.0 Hz, 2 H), 7.60 (t, *J =* 7.2 Hz, 1 H), 7.50 (t, *J =* 8.0 Hz, 2 H), 7.40 (t, *J =* 7.0 Hz, 2 H), 7.35–7.27 (m, 3 H), 6.26 (t, *J =* 8.6 Hz, 1 H), 5.65 (d, *J =* 7.1 Hz, 1 H), 5.47 (br s, 1 H), 5.41–5.29 (m, 3 H), 5.00 (t, *J =* 7.0 Hz, 2 H), 4.31 (d, *J =* 8.8 Hz, 1 H), 4.17 (d, *J =* 8.1 Hz, 1 H), 3.89 (dd, *J* = 11.1, 6.6 Hz, 1 H), 3.73 (s, 3 H), 3.44 (s, 3 H), 3.30 (s, 3 H), 2.72 (t, *J =* 7.0 Hz, 3 H), 2.64–2.48 (m, 5 H), 2.44 (s, 3 H), 2.36–2.25 (m, 1 H), 2.23–2.13 (m, 1 H), 2.08–1.93 (m, 10 H), 1.86–1.76 (m, 3 H), 1.72 (s, 3 H), 1.61 (s, 6 H), 1.43–1.37 (m, 2 H), 1.35 (s, 9 H), 1.33–1.18 (m, 24 H), 0.88 (t, *J =* 6.8 Hz, 3 H); ^13^C NMR (101 MHz, CDCl_3_) *δ* 205.1, 172.7, 172.1, 170.9, 169.8, 168.4, 167.2, 139.6, 135.1, 133.7, 130.3, 130.2, 129.7, 129.4, 129.0, 128.8, 128.3, 126.5, 84.3, 82.6, 81.7, 80.8, 80.6, 79.0, 76.6, 74.9, 74.6, 72.5, 72.2, 57.3, 57.2, 56.9, 52.3, 47.5, 43.5, 37.6, 37.2, 35.1, 32.4, 32.1, 32.0, 31.2, 29.9, 29.7, 29.4, 29.2, 29.1, 28.3, 27.4, 27.2, 26.8, 25.2, 24.2, 24.0, 22.9, 22.8, 21.1, 14.6, 14.2, 10.5; ESI-MS (*m/z*) [M+Na] calculated for C_72_H_103_NO_19_S_2_ 1372.65. Found 1372.56.

**18a**: White foam; Reaction time = 1.5 h; Yield = 88% (45.3 mg, 0.035 mmol); R_f_ (1:1 *n*-hexane/EtOAc) = 0.2; ^1^H NMR (400 MHz, CDCl_3_) δ ^1^H NMR (400 MHz, CDCl_3_) δ 8.10 (d, *J* = 7.3 Hz, 2 H), 7.60 (t, *J* = 7.5 Hz, 1 H), 7.49 (t, *J* = 7.8 Hz, 2 H), 7.38 (t, *J* = 7.4 Hz, 2 H), 7.29 (t, *J* = 8.4 Hz, 3 H), 6.24 (t, *J* = 8.5 Hz, 1 H), 5.67 (d, *J* = 7.1 Hz, 1 H), 5.51–5.28 (m, 5 H), 5.21 (s, 1 H), 4.97 (q, *J* = 6.5 Hz, 2 H), 4.31 (d, *J* = 8.4 Hz, 1 H), 4.27 (dd, *J* = 11.0, 6.5 Hz, 1 H), 4.19 (d, *J* = 8.4 Hz, 1 H), 3.92 (d, *J* = 7.0 Hz, 1 H), 3.73 (s, 3 H), 2.62–2.51 (m, 1 H), 2.43 (s, 3 H), 2.41–2.24 (m, 5 H), 2.19–2.09 (m, 1 H), 2.05–1.96 (m, 4 H), 1.94 (s, 3 H), 1.88–1.77 (m, 4 H), 1.74 (s, 4 H), 1.63 (quintet, *J* = 7.2 Hz, 2 H), 1.51 (quintet, *J* = 7.2 Hz, 2 H), 1.41–1.33 (m, 3 H), 1.32 (s, 9 H), 1.31–1.17 (m, 29 H), 1.11 (s, 3 H), 0.87 (t, *J* = 6.7 Hz, 3 H); ^13^C NMR (101 MHz, CDCl_3_) δ 211.1, 173.6, 172.9, 171.1, 169.8, 168.3, 167.2, 139.2, 135.6, 133.7, 130.3, 130.2, 129.7, 129.4, 128.9, 128.8, 128.2, 126.4, 84.4, 81.1, 79.0, 76.7, 75.1, 74.6, 74.2, 72.3, 72.0, 71.9, 57.7, 52.3, 46.5, 43.2, 37.0, 35.8, 34.1, 33.8, 32.0, 31.2, 29.9, 29.7, 29.6, 29.4, 29.4, 29.2, 29.1, 29.0, 28.9, 28.3, 27.3, 27.2, 26.4, 25.2, 24.9, 24.7, 22.8, 22.8, 21.0, 14.3, 14.2, 10.1; ESI-MS (*m*/*z*) [M+Na] calculated for C_72_H_103_NO_19_ 1308.70. Found 1309.42.

**18b**: White foam; Reaction time = 1.5 h; Yield = 69% (36.4 mg, 0.028 mmol); R_f_ (1:1 *n*-hexane/EtOAc) = 0.16; ^1^H NMR (400 MHz, CDCl_3_) *δ* 8.11 (d, *J =* 7.2 Hz, 2 H), 7.60 (t, *J =* 7.4 Hz, 1 H), 7.50 (t, *J =* 8.0 Hz, 2 H), 7.39 (t, *J =* 7.2 Hz, 2 H), 7.34–7.27 (m, 3 H), 6.24 (t, *J =* 9.8 Hz, 1 H), 5.68 (d, *J =* 7.1 Hz, 1 H), 5.47 (s, 2 H), 5.41–5.29 (m, 3 H), 5.22 (s, 1 H), 4.97 (q, *J =* 6.5 Hz, 2 H), 4.32 (d, *J =* 8.4 Hz, 1 H), 4.26 (dd, *J* = 11.0, 6.5 Hz, 1 H), 4.19 (d, *J =* 8.4 Hz, 1 H), 3.92 (d, *J =* 7.1 Hz, 1 H), 3.72 (d, *J =* 2.6 Hz, 3 H), 2.71 (t, *J =* 7.1 Hz, 2 H), 2.65–2.45 (m, 7 H), 2.44 (s, 3 H), 2.37–2.27 (m, 1 H), 2.20–2.11 (m, 1 H), 2.08–1.95 (m, 7 H), 1.94 (s, 4 H), 1.89–1.77 (m, 5 H), 1.74 (s, 3 H), 1.413–1.35 (m, 3 H), 1.33 (s, 9 H), 1.31–1.24 (m, 18 H), 1.23 (s, 3 H), 1.12 (s, 3 H), 0.88 (t, *J =* 6.7 Hz, 3 H); ^13^C NMR (101 MHz, CDCl_3_) *δ* 211.6, 172.8, 172.1, 171.0, 169.8, 168.3, 167.2, 155.3, 139.2, 135.7, 133.8, 130.3, 130.2, 129.7, 129.4, 129.0, 128.8, 128.3, 126.4, 84.3, 81.1, 80.5, 79.0, 76.7, 75.1, 74.6, 74.5, 72.5, 72.1, 72.0, 57.7, 52.4, 46.5, 43.2, 37.6, 37.2, 35.7, 32.4, 32.0, 31.2, 29.9, 29.7, 29.6, 29.4, 29.1, 28.3, 27.4, 27.3, 26.4, 25.2, 24.2, 24.0, 21.0, 14.3, 10.1; ESI-MS (*m/z*) [M+Na] calculated for C_70_H_99_NO_19_ 1344.62. Found 1345.12.

**19a**: Yellow solid; Reaction time = 24 h; Yield = 95% (31.4 mg, 0.038 mmol); R_f_ (2:8 *n*-hexane/EtOAc) = 0.25; ^1^H NMR (400 MHz, CDCl_3_) *δ* 8.40 (s, 1 H), 8.22 (d, *J =* 8.6 Hz, 1 H), 7.49 (d, *J =* 8.2 Hz, 1 H), 7.87–7.81 (m, 1 H), 7.69–7.64 (m, 1 H), 7.23 (br s, 1 H), 5.67 (t, *J =* 17.2 Hz, 1 H), 5.41 (d, *J =* 17.2 Hz, 1 H), 5.37–5.30 (m, 2 H), 5.30 (s, 2 H), 4.96 (t, *J =* 6.6 Hz, 1 H), 3.72 (s, 3 H), 2.54–2.40 (m, 2 H), 2.34–2.24 (m, 3 H), 2.20–2.10 (m, 1 H), 2.04–1.96 (m, 3 H), 1.84–1.76 (m, 2 H), 1.70–1.50 (m, 8 H), 1.39–1.23 (m, 25 H), 0.97 (t, *J =* 7.6 Hz, 3 H), 0.87 (t, *J =* 6.8 Hz, 3 H); ^13^C NMR (101 MHz, CDCl_3_) *δ* 173.45, 172.9, 171.1, 167.7, 157.5, 152.5, 148.9, 146.2, 146.1, 131.4, 130.8, 130.2, 129.7, 129.7, 128.6, 128.4, 128.3, 128.2, 120.5, 96.2, 75.7, 72.2, 67.2, 52.3, 50.0, 34.0, 33.9, 32.0, 31.9, 31.2, 29.9, 29.7, 29.6, 29.4, 29.4, 29.2, 29.1, 29.1, 29.1, 29.0, 27.3, 27.2, 25.7, 25.2, 25.0, 24.8, 24.7, 22.8, 14.2, 7.7; ESI-MS (*m/z*) [M+Na], [2M+Na] calculated for C_49_H_66_N_2_O_9_ 849.47, 1675.94. Found 849.69, 1675.11.

**19b**: Yellow solid; Reaction time = 24 h; Yield = 73% (25.2 mg, 0.029 mmol); R_f_ (2:8 *n*-hexane/EtOAc) = 0.22; ^1^H NMR (400 MHz, CDCl_3_) *δ* 8.40 (s, 1 H), 8.24 (d, *J =* 8.4 Hz, 1 H), 7.94 (d, *J =* 8.2 Hz, 1 H), 7.88–7.79 (m, 1 H), 7.70–7.63 (m, 1 H), 7.23 (s, 1 H), 5.68 (t, *J =* 17.2 Hz, 1 H), 5.41 (d, *J =* 17.2 Hz, 1 H), 5.33 (sextet, *J =* 5.1 Hz, 2 H), 5.29 (s, 2 H), 4.96 (t, *J =* 6.5 Hz, 1 H), 3.72 (s, 3 H), 2.75–2.58 (m, 6 H), 2.52–2.43 (m, 2 H), 2.83 (sextet, *J =* 7.4 Hz, 1 H), 2.16 (quintet, *J =* 7.6 Hz, 1 H), 2.09–1.96 (m, 8 H), 1.83–1.76 (m, 2 H), 1.39–1.21 (m, 21 H), 0.98 (t, *J =* 7.4 Hz, 3 H), 0.87 (t, *J =* 6.8 Hz, 3 H); ^13^C NMR (101 MHz, CDCl_3_) *δ* 172.6, 172.1, 170.9, 167.6, 157.5, 152.4, 148.9, 146.3, 146.0, 131.4, 130.9, 130.2, 129.7, 129.7, 128.6, 128.3, 128.3, 128.2, 120.4, 96.2, 76.1, 72.4, 67.1, 52.3, 50.1, 37.5, 37.3, 33.9, 32.4, 32.3, 31.9, 31.2, 29.9, 29.7, 29.6, 29.4, 29.4, 29.2, 29.1, 27.3, 27.2, 25.7, 25.2, 25.0, 24.2, 24.0, 22.8, 14.2, 7.7; ESI-MS (*m/z*) [M+Na], [2M+Na] calculated for C_47_H_62_N_2_O_9_S_2_ 885.38, 1747.77. Found 885.53, 1747.74.

**20a**: Colorless oil; Reaction time = 22 h; Yield = 36% (14 mg, 0.014 mmol); R_f_ (1:1 *n*-hexane/EtOAc) = 0.29; ^1^H NMR (400 MHz, CDCl_3_) *δ* 6.98 (s, 1 H), 6.62 (s, 1 H), 5.46 (d, *J =* 7.4 Hz, 1 H), 5.37–5.27 (m, 3 H), 4.97 (t, *J =* 6.5 Hz, 1 H), 4.13–4.04 (m, 1 H), 3.72 (s, 3 H), 3.36–3.30 (m, 1 H), 3.07–3.02 (m, 1 H), 2.90–2.85 (m, 1 H), 2.70 (s, 3 H), 2.53–2.50 (m, 1 H), 2.40–2.28 (m, 5 H), 2.10 (s, 3 H), 2.03–1.96 (m, 4 H), 1.84–1.76 (m, 2 H), 1.68–1.58 (m, 7 H), 1.35–1.22 (m, 38 H), 1.09 (s, 3 H), 1.03 (d, *J =* 6.8 Hz, 3 H), 0.87 (t, *J =* 6.8 Hz, 6 H); ^13^C NMR (101 MHz, CDCl_3_) *δ* 216.6, 173.5, 173.5, 171.1, 170.7, 130.2, 129.7, 117.4, 116.5, 78.4, 76.1, 72.2, 58.2, 53.6, 52.5, 52.3, 49.6, 44.0, 38.5, 34.5, 34.4, 34.0, 32.0, 31.6, 31.2, 29.9, 29.7, 29.6, 29.4, 29.4, 29.3, 29.2, 29.1, 29.1, 27.3, 27.2, 26.4, 25.7, 25.2, 25.2, 25.1, 24.9, 24.2, 22.8, 17.7, 15.7, 14.2; ESI-MS (*m/z*) [M+Na] calculated for C_55_H_89_NO_11_S 994.61. Found 995.54.

**20b**: Colorless oil; Reaction time = 19 h; Yield = 55% (22.2 mg, 0.022 mmol); R_f_ (1:1 *n*-hexane/EtOAc) = 0.24; ^1^H NMR (400 MHz, CDCl_3_) δ 6.97 (s, 1 H), 6.62 (s, 1 H), 5.46 (dd, *J* = 7.2, 3.0 Hz, 1 H), 5.37–5.28 (m, 4 H), 4.97 (t, *J* = 6.4 Hz, 1 H), 4.11–4.07 (m, 1 H), 3.72 (s, 3 H), 3.36–3.30 (m, 1 H), 3.07–3.02 (m, 1 H), 2.90–2.85 (m, 1 H), 2.75–2.70 (m, 4 H), 2.69 (s, 3 H), 2.54–2.47 (m, 5 H), 2.09 (s, 3 H), 2.06–2.00 (m, 7 H), 1.84–1.76 (m, 2 H), 1.62–1.47 (m, 4 H), 1.37–1.22 (m, 31 H), 1.09 (s, 3 H), 1.03 (d, *J* = 6.8 Hz, 3 H), 0.87 (t, *J* = 7.1 Hz, 6 H); ^13^C NMR (101 MHz, CDCl_3_) δ 216.6, 176.9, 172.6, 172.6, 170.9, 170.7, 130.2, 129.7, 116.4, 78.6, 76.1, 74.1, 72.4, 57.6, 54.1, 52.4, 52.3, 43.8, 38.5, 37.8, 37.7, 37.7, 37.6, 34.4, 32.7, 32.4, 32.3, 31.9, 31.1, 29.8, 29.7, 29.6, 29.4, 29.1, 29.1, 27.3, 27.2, 26.3, 25.2, 24.4, 24.2, 24.2, 24.1, 22.8, 19.1, 17.7, 15.8, 15.6, 14.2; ESI-MS (*m*/*z*) [M+Na] calculated for C_53_H_85_NO_11_S_3_ 1030.52. Found 1031.48.

**21a**: Colorless oil; Reaction time = 40 h; Yield = 66% (23.5 mg, 0.026 mmol); R_f_ (1:1 *n*-hexane/EtOAc) = 0.23; ^1^H NMR (400 MHz, CDCl_3_) *δ* 6.70 (s, 1 H), 6.48 (s, 1 H), 6.34 (s, 2 H), 5.92 (d, *J* = 5.2 Hz, 2 H), 5.84 (d, *J* =9.0 Hz, 1 H), 5.34–5.25 (m, 2 H), 4.93 (t, *J* =6.6 Hz, 1 H), 4.53 (d, *J* =4.4 Hz, 1 H), 4.30 (t, *J* = 9.3 Hz, 1 H), 4.15 (t, *J* = 10.0 Hz, 1 H), 3.72 (s, 3 H), 3.71 (s, 6 H), 3.68 (s, 3 H), 2.87 (dd, *J* =14.5, 4.4 Hz 1 H), 2.42–2.29 (m, 4 H), 2.00–1.91 (m, 4 H), 1.81–1.73 (m, 2 H), 1.67–1.55 (m, 4 H), 1.36–1.16 (m, 31 H), 0.83 (t, *J* = 6.7 Hz, 3 H); ^13^C NMR (101 MHz, CDCl_3_) *δ* 174.1, 173.7, 173.3, 170.9, 152.6, 148.1, 147.6, 137.0, 134.9, 132.2, 130.0, 129.6, 128.5, 109.7, 108.0, 107.0, 101.6, 73.3, 72.1, 71.4, 60.7, 56.1, 52.1, 45.5, 43.7, 38.7, 34.3, 33.8, 31.9, 31.1, 31.0, 29.6, 29.3, 29.0, 27.2, 27.1, 25.1, 24.9, 22.6, 14.1; ESI-MS (*m/z*) [M+Na], [2M+Na] calculated for C_51_H_72_O_13_ 915.49, 1807.98. Found 916.08, 1807.03.

**21b**: Colorless oil; Reaction time = 46 h; Yield = 75% (23.5 mg, 0.025 mmol); R_f_ (1:1 *n*-hexane/EtOAc) = 0.19; ^1^H NMR (400 MHz, CDCl_3_) δ 6.73 (s, 1 H), 6.49 (s, 1 H), 6.35 (s, 2 H), 5.94 (d, *J* = 4.9 Hz, 1 H), 5.85 (d, *J* = 9.0 Hz, 1 H), 5.36–5.24 (m, 2 H), 4.95 (t, *J* = 6.5 Hz, 1 H), 4.55 (d, *J* =4.4 Hz, 1 H), 4.33 (t, *J* = 9.4 Hz, 1 H), 4.16 (t, *J* = 10.0 Hz, 1 H), 3.76 (s, 3 H), 3.72 (s, 6 H), 3.69 (s, 3 H), 2.89 (dd, *J* = 14.4, 4.4 Hz, 1 H), 2.75–2.67 (m, 4 H), 2.58–2.43 (m, 4 H), 2.11–1.91 (m, 8 H), 1.82–1.73 (m, 2 H), 1.39–1.13 (m, 24 H), 0.84 (t, *J* = 6.7 Hz, 3 H); ^13^C NMR (101 MHz, CDCl_3_) δ 173.7, 173.4, 172.5, 170.8, 152.6, 148.1, 147.6, 137.0, 134.8, 132.3, 130.1, 129.6, 128.3, 109.7, 107.9, 106.9, 101.6, 73.7, 72.3, 71.4, 60.7, 56.1, 52.2, 45.2, 43.7, 38.7, 37.5, 37.4, 32.5, 32.2, 31.9, 31.0, 29.7, 29.3, 29.0, 27.2, 27.1, 25.1, 24.1, 23.9, 22.7, 14.1; ESI-MS (*m*/*z*) [M+Na] calculated for C_49_H_68_O_13_S_2_ 951.40. Found 951.46.

**22a**: 0.8 mmol triethylamine (Et_3_N) was also added at 0 °C for its preparation; Light yellow oil; Reaction time = 22 h; Yield = 86% (29.3 mg, 0.034 mmol); R_f_ (pure EtOAc) = 0.58; ^1^H NMR (300 MHz, CDCl_3_) δ 7.30 (s, 1 H), 7.25–7.18 (m, 1 H), 7.06 (d, *J* = 10.5 Hz, 1 H), 6.53 (s, 1 H), 6.17 (br s, 1 H), 5.41–5.29 (m, 2 H), 5.12–4.93 (m, 1 H), 4.71–4.61 (m, 1 H), 3.94 (s, 3 H), 3.90 (s, 3 H), 3.73 (s, 3 H), 3.66 (s, 3 H), 2.43 (s, 3 H), 2.39-2.32 (m, 4 H), 2.24–2.16 (m, 4 H), 2.11–1.94 (m, 4 H), 1.84–1.79 (m, 2 H), 1.62–1.60 (m, 4 H), 1.30 (m, 27 H), 0.88 (t, *J* = 6.6 Hz, 3 H); ^13^C NMR (75 MHz, CDCl_3_) δ 182.4, 173.3, 173.2, 170.9, 158.1, 153.6, 152.0, 151.2, 141.6, 138.7, 134.7, 134.4, 130.0, 129.6, 128.7, 126.7, 125.7, 107.4, 72.0, 61.6, 61.3, 56.0, 52.1, 51.8, 36.7, 36.2, 33.9, 31.8, 31.0, 30.0, 29.5, 29.2, 28.9, 27.2, 27.1, 25.5, 25.0, 24.8, 22.6, 15.0, 14.1; ESI-MS (*m*/*z*) [M+Na], [2M+Na] calculated for C_49_H_73_NO_9_S 874.49, 1725.99. Found 874.78, 1724.78.

**22b**: 0.8 mmol triethylamine (Et_3_N) was also added at 0 °C for its preparation; Light yellow oil; Reaction time = 46 h; Yield = 72% (25.5 mg, 0.029 mmol); R_f_ (pure EtOAc) = 0.54; ^1^H NMR (400 MHz, CDCl_3_) *δ* 7.59 (s, 1 H), 7.45 (d, *J* = 10.6 Hz, 2 H), 7.21 (d, *J* = 10.6 Hz, 2 H), 6.90 (s, 1 H), 5.39–5.31 (m, 1 H), 4.98 (t, *J* = 6.5 Hz, 1 H), 4.78–4.62 (m, 1 H), 3.92 (d, *J* = 12.9 Hz, 6 H), 3.73 (s, 3 H), 3.66 (s, 3 H), 2.76–2.62 (m, 3 H), 2.59–2.45 (m, 6 H), 2.44–2.27 (m, 8 H), 2.10–1.90 (m, 7 H), 1.85–1.76 (m, 2 H), 1.56 (br s, 1 H), 1.45–1.17 (m, 20 H), 0.88 (t, *J* = 6.6 Hz, 3 H); ^13^C NMR (101 MHz, CDCl_3_) *δ* 181.2, 172.9, 171.9, 171.0, 167.1, 158.8, 154.0, 151.3, 145.3, 141.9, 136.0, 134.4, 130.4, 129.9, 128.2, 127.9, 125.4, 107.6, 72.5, 61.9, 61.5, 56.3, 52.4, 38.1, 37.7, 37.2, 34.6, 32.6, 32.0, 31.2, 30.1, 29.9, 29.7, 29.5, 29.2, 27.4, 27.3, 25.3, 24.6, 24.3, 22.9, 15.4, 14.2; ESI-MS (*m/z*) [M+Na], [2M+Na] calculated for C_47_H_69_NO_9_S_3_ 910.40, 1797.82. Found 910.87, 1796.93.

#### 3.2.5. Deacetylation of Thiocolchicine **13**

To a solution of (–)-thiocolchicine **13** (500 mg, 1.2 mmol) in MeOH (20 mL), aqueous 2N HCl (9.65 mL, 19.3 mmol) was added, and the reaction mixture was stirred at reflux for 48 h. MeOH was then removed under reduced pressure, H_2_O (20 mL) was added, and the resulting solution was extracted with CH_2_Cl_2_ (3 × 15 mL). The aqueous layer was neutralized with aqueous 1N NaOH and extracted with CH_2_CI_2_ (3 × 15 mL). The combined organic phases were washed with brine (20 mL), dried over Na_2_SO_4,_ and concentrated under reduced pressure. The residue thus obtained was purified by direct FCC (9:1 CH_2_Cl_2_/MeOH) to obtain pure target **23** as a pale-yellow solid (387 mg, 1.02 mmol, 86%). ^1^H NMR (300 MHz, CDCl_3_) *δ* 7.61 (s, 1 H), 7.22 (d, *J* = 10.5 Hz, 1 H), 7.05 (d, *J* = 10.5 Hz, 1 H), 6.56 (s, 1 H), 3.93 (s, 6 H), 3.83–3.79 (m, 1 H), 3.69 (s, 3 H), 2.54–2.50 (m, 1 H), 2.47–2.38 (m, 5 H), 1.79–1.93 (m, 1 H); ESI-MS (*m/z*) [M+H] calculated for C_20_H_23_NO_4_S 374.14. Found 374.26.

### 3.3. Nanoparticles Preparation

Nanoparticle formulations **16**–**22a,bNPs** were prepared by the solvent displacement method [35]. In detail, dru conjugates **16**–**22a,b** containing methyl 2-hydroxy oleate were dissolved in EtOH (4 mg/mL), and the obtained solution was diluted up to 1.3 mg/mL by adding it dropwise at room temperature to ultrapure sterile water under vigorous stirring. Finally, the organic solvent was evaporated under reduced pressure at 40 °C to obtain a final 2 mg/mL aqueous suspension of formulations **16**–**22a,bNPs**.

### 3.4. Dynamic Light Scattering (DLS)

DLS measurements were carried out by a 90-plus particle size analyzer (Brookhaven Instruments Corporation, Holtsville, NY, USA) equipped with a solid state He−Ne laser (wavelength = 661 nm). Experiments were carried out at a scattering angle of 90° on samples at 298 K. For both DLS and Z-potential analysis, purified samples were diluted in distilled water to a concentration of 200 µg/mL and briefly sonicated prior to the analysis. Results were expressed as mean ± standard deviation (SD) of three measurements.

### 3.5. Nanoparticles Characterization by TEM

Transmission electron microscopy (TEM) measurements were performed with a ZEISS LIBRA200FE microscope equipped with in column Ω-filter spectrometer/filter, operating at 200 kV. TEM specimens were prepared by dropping the nanoparticle dispersion onto a supported ultrathin-carbon film copper TEM grids and analyzed after drying overnight. The dimensions of NPs have been measured using the ITEM imaging platform—Olympus Soft Imaging Solutions

### 3.6. Cell Cultures

MSTO-211H (human biphasic mesothelioma), HT-29 (human colorectal adenocarcinoma), and Met-5A (human mesothelium) cells were grown in RPMI 1640 (R6504, Sigma Chemical Co.) supplemented with 10% heat-inactivated fetal calf serum (FCS) (F7524, Sigma Chemical Co.). For MSTO-211H and Met-5A cells, 2.38 g/L Hepes, 0.11 g/L pyruvate sodium, and 2.5 g/L glucose were added to the medium. LN229 (human glioblastoma) cells were cultured in DMEM (D2902, Sigma Chemical Co.) supplemented with 3.5 g/L glucose and 5% FCS.

100 U/mL penicillin, 100 μg/mL streptomycin, and 0.25 μg/mL amphotericin B (Sigma Chemical Co.) were added to all media. Cells were cultured in a humidified atmosphere incubator containing 5% carbon dioxide in air at 37 °C.

### 3.7. Inhibition Growth Assay

Trypan blue staining was performed to assess cell viability. Cells (3–3.5 × 10^4^) were seeded into each well of a 24-well cell culture plate. After incubation for 24 h in standard conditions, various concentrations of nanoformulations **15**–**22a,bNPs** or free drugs **7**–**13** were added, and cells were incubated for a further 72 h. Cytotoxicity data were expressed as GI_50_ values, that is, the concentration of the added agent able to induce a 50% reduction in cell number with respect to a control untreated culture.

### 3.8. Intracellular Reactive Oxygen Species (ROS) Measurement

To quantify intracellular ROS determination, 2′,7′-dichlorofluorescein diacetate (D6883, Sigma-Aldrich Chemical Co.) was used as a fluorogenic probe. LN229 cells (6 × 10^3^/well) were seeded in 200 µL complete medium into a 96-well cell culture plate and allowed to grow for 48 h. We added 3 mM N-acetylcysteine (NAC, A7250 Sigma-Aldrich Chemical Co., St. Louis, MO, USA) for 30 min in the complete medium. Then, the medium was discarded, and cells were washed with phosphate-buffered saline (8 mM Na_2_HPO_4_·2H_2_O, 1.5 mM KH_2_PO_4_, 2 mM KCl, 0.1 M NaCl, PBS) and incubated with 10 µM 2′,7′-dichlorofluorescein diacetate in PBS-glucose 5 mM at 37 °C in the dark for 20 min. After incubation, the solution was removed, cells were washed with PBS, and either **17bNP** or cabazitaxel **8** were added at various concentrations in 5 mM PBS-glucose. Fluorescence was detected for 30 min by a microplate reader (Victor X3 Multilabel plate reader, Perkin Elmer) at λex = 485 nm and λem = 527 nm. ROS production was calculated at 30 min after subtracting autofluorescence value (cells without 2′,7′-dichlorofluorescein diacetate addition) with respect to untreated cells (control).

### 3.9. Cell Cycle Analysis

LN229 cells (5 × 10^5^) were seeded in culture plates with complete medium, and after 24 h, they were treated with 6 nM **9**, 100 nM **12**, 400 nM **18bNP,** and 100 μM **21bNP** and incubated for a further 24 h in standard conditions. Cells were harvested, centrifuged, and treated with ice-cold 70% *w*/*v* ethanol at 4 °C for 20 min. Then, cells were washed twice with phosphate buffer saline (PBS) and resuspended in a final volume of 300 μL of PBS containing 0.1 μg mL^−1^ RNAse (Merck R6513) and 36 μg mL^−1^ propidium iodide (PI, Merck P4170). The analysis of the DNA content was performed by FACSAria III flow cytometry, and the data were analyzed by BD FACSDiva software.

## 4. Conclusions

We used 2-hydroxyoleic acid 2OHOA as a building block for the preparation of 14 drug-conjugates **16**–**22a,b** containing one among seven anticancer drugs **7**–**13**. All drug conjugates were capable of self-assembly, leading to nanoparticles **16**–**22a,bNPs**. Their biological evaluation evidenced relevant cytotoxicity towards three human cancer cell lines (MSTO-211H—biphasic mesothelioma, HT-29—colorectal adenocarcinoma, and LN-229—glioblastoma), with GI_50_ in the low micromolar range. Nanoparticles **16**–**22bNPs** bearing a disulfide bond in their linker showed a higher antiproliferative activity, which seems to confirm its desired beneficial effect in the release of a free drug with the subsequent preservation of its molecular mechanism of action, as demonstrated for the most active nanoformulations **17bNP** and **18bNP**, and from **21bNP**.

## Data Availability

Data is contained within the article and Appendix A.

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
