# Peer review of "2-Hydroxyoleic Acid as a Self-Assembly Inducer for Anti-Cancer Drug-Centered Nanoparticles"

_pharmaceuticals, 2023, doi:10.3390/ph16050722_

Round 1
Reviewer 1 Report
Comment for pharmaceuticals-2327400
This manuscript was submitted by L. Dalla Via et al and co-authors, “2-Hydroxyoleic acid as a self-assembly inducer for anti-cancer drug-centered nanoparticles”. The authors presented the details in the manuscript in a good format. Nevertheless, there are still some issues needed to be addressed before publication in this journal.
1. In the introduction, the author needs to elaborate on the disulfide-crosslinked role on drug conjugated nanoparticles.
2. The author needs to discuss the in vitro drug release of disulfide-crosslinker-based nanoparticles and incorporate the results and the author needs to explain the role of disulfide-crosslinker.
3. The author needs to provide the TEM or SEM of nanoparticles.
4. In the NMR figures, there is no peak assignment of each molecule, so the authors need to include the peak assignment of the molecules in the NMR figure.
5. Please check for grammar corrections.

Author Response
- In the introduction, the author needs to elaborate on the disulfide-crosslinked role on drug conjugated nanoparticles.
- The author needs to discuss the in vitro drug release of disulfide-crosslinker-based nanoparticles and incorporate the results and the author needs to explain the role of disulfide-crosslinker.
Ans.: The following paragraph together with the corresponding bibliography was added to the introduction; lines 82-89:
“The presence of the disulfide bond is crucial for the triggered drug release at the tumor side, as the disulfide bond is stable under physiological body temperature, pH, and oxidation environment, whereas it can be degraded by reducing agents such as glutathione (GSH). In particular, the intracellular concentration of GSH is much higher than the extracellular one, due to the high amount of GSH produced by cancer cells in comparison to normal cells [22-24]. Thus, the disulfide based crosslinking approach was considered as the most suitable for the preparation of our conjugates.”
- The author needs to provide the TEM or SEM of nanoparticles.
Ans.: TEM analysis was performed and the results were added to the manuscript; see Figure 5.
- In the NMR figures, there is no peak assignment of each molecule, so the authors need to include the peak assignment of the molecules in the NMR figure.
Ans.: The NMR of the drugs used on this work are reported in literature. Moreover, according to 1-D 1NMR the number of protons represented by the signal, peak multiplicity, and coupling constants have been reported. Peak assignment is reported when 2-D experiments have been performed.
- Please check for grammar corrections.
Ans.: Grammar check was performed.
Reviewer 2 Report
The authors present an interesting study about the synthesis of a new nanoparticle family based on the self-assembly of 2-hydroxyoleic acid. The aim is of interest but there are several points to be addressed.
1. Although the chemical properties of the synthesized molecules are well defined, the nanoparticles characterization is mainly based on size, size distribution and Z-potential determinations. Additional measurements of morphology should be performed.
2. Strictly speaking, nanoparticles are considered structures with one dimension less of 100 nm. Therefore, the assessment of lines 135-137 refutable. The internalization of structures above 200 nm is also questionable.
3. Cells. Control cellular lines are missing.
4. Figures. The authors have prioritized figures about the chemistry of the synthesized compounds that is of interest for the manuscript, however, additional figures such as those illustrating the cytotoxicity tests, would enrich the manuscript.
Author Response
- Although the chemical properties of the synthesized molecules are well defined, the nanoparticles characterization is mainly based on size, size distribution and Z-potential determinations. Additional measurements of morphology should be performed.
Ans.: TEM analysis was performed and the results were added to the manuscript; see Figure 5.
- Strictly speaking, nanoparticles are considered structures with one dimension less of 100 nm. Therefore, the assessment of lines 135-137 refutable. The internalization of structures above 200 nm is also questionable.
Ans.: Our group has monitored in the past [reference 4: Fumagalli et al. ChemPlusChem. 2015, 80, 1380-1383, doi:10.1002/cplu.201500156] the ability of hetero nanoparticles [mean diameter: (224±2) nm].to pass through the cell membrane in biological media, using direct stochastic optical reconstruction microscopy (dSTORM), providing the following results.
“Breast cancer MCF-7 cells were treated with the fluorescent hetero-nanoparticles for 20 min and imaged by multiple fluorescence microscopy methods. Three-dimensional confocal microscopy allowed the identification of the fluorescent nanoparticles in the cytoplasm of the cells, providing evidence that the nanoparticles are capable of permeating the cellular membrane. However, to provide an estimation of the size of the NPs, the resolution of conventional microscopy is not sufficient. We therefore applied super-resolution microscopy (dSTORM) to quantify the diameter of the bright particles detected inside the cells. The enlarged detail of the NP served to quantify its diameter: 156 nm. The diameter is in good agreement with our previous measurements by QELS [(224±2) nm]. In order to monitor the behaviour of the internalized NPs, we performed dStorm imaging at different incubation times. In this way we confirmed a progressive increase in the variability of the nanoparticle diameter—as indicated by the increase in the coefficient of variation—which might be an indication of their tendency to disassemble.”
- Cells. Control cellular lines are missing.
Ans.: The cytotoxicity data of the non-tumorigenic Met-5A (mesothelial) cells on the most effective NPs (17bNP and 18bNP) and on the corresponding free drugs (cabazitaxel (8) and docetaxel (9)), are reported as control, in the revised manuscript version (lines 210-216).
- Figures. The authors have prioritized figures about the chemistry of the synthesized compounds that is of interest for the manuscript, however, additional figures such as those illustrating the cytotoxicity tests, would enrich the manuscript.
Ans.: As suggested by the reviewer, some figures illustrating the dose-dependent behavior in the cytotoxicity tests are inserted in the revised manuscript. The new figures 6A and B show the obtained results of treatments in presence of 8, 9 and the corresponding NPs (17aNP, 17bNp, 18aNP and 18bNP). The graphics regarding other tested compounds and NPs are added in Supplementary Information (Figure S37, SI).
Reviewer 3 Report
The presented manuscript covers synthesis and cytotoxicity evaluation of 2OHOA nanoparticles in cancer cell lines. In this manuscript, the authors synthesized a series of 2OHOA nanoparticles, characterized with mass spectrum and DLS, and assess GI50 on three different cell lines. The authors proved that proposed NPs exhibit antiproliferative activity and were able to induce ROS production. There are several questions/suggestions I have.
1. The major concern I have is the amount of data provided in this manuscript may be inadequate for this journal. The authors should either provide in vivo data or more in vitro MOA studies.
2. The authors need to provide all the results (mass spectrum/NMR and cytotoxicity curves) in manuscript or in supplemental files.
3. The organization of this manuscript is not proper. There is no Section 3. Also, a lot of details were missing. For example, the author mentioned “good yield” or “excellent yield” , what’s the number of the yield?
4. Missing control groups. It’s better to include free drug and free drug linker treatment control group in the cytotoxicity assay to prove the antiproliferative activity is due to NPs.
5. The sizes of the NPs are all above 100 nm, some of them exceeds 300 nm, which means most of the NPs will be recognized by mononuclear phagocytes, preventing them from entering other tissues. Those NPs will be mostly in liver, spleen and blood. How is the cytotoxicity of these NPs in liver?
6. If the authors haven’t performed the in vivo study, is there any proof from literature indicating the efficacy and safety of such NPs?
Author Response
- The major concern I have is the amount of data provided in this manuscript may be inadequate for this journal. The authors should either provide in vivo data or more in vitro MOA studies.
Ans.: The in vivo experiments are usually carried out for agents that are very promising in view of human therapeutic use, and in Italy the ethic indications are very strict, so that the permission for in vivo analysis requires usually long time to be attained and convincing preliminary data that could justify the treatment of animals. In our opinion, the proposed NPs are worthy to be considered by the scientific world because they represent an increase in knowledge, and confirm an interesting approach in drug delivery, but the obtained results appear still quite far from the requirements necessary to obtain the in vivo approval from our ethical committee.
For the above reasons we did not perform such experiments. Nevertheless, as required, we performed further MoA studies, by investigating the effect of conjugates 18bNP and 21bNP and free drugs docetaxel (9) and podophyllotoxin (12) on cell cycle. The obtained results are shown in Figure 8A,B of the revised manuscript.
- The authors need to provide all the results (mass spectrum/NMR and cytotoxicity curves) in manuscript or in supplemental files.
Ans.: As required, cytotoxicity curves of drugs and conjugates are reported in the Supplementary section. The results regarding 8, 9 and the corresponding NPs (17aNP, 17bNP, 18aNP and 18bNP) are shown in Figure 6A,B of the revised manuscript.
- The organization of this manuscript is not proper. There is no Section 3. Also, a lot of details were missing. For example, the author mentioned “good yield” or “excellent yield”, what’s the number of the yield?
Ans.: Section’s numbering was corrected. Section 4., 4.1 etc is now 3., 3.1 etc..
Below every scheme the yield or yield range is now reported. Furthermore, in Section 3.2: Experimental procedures there is the yield for every single synthesized intermediate or target compound.
- Missing control groups. It’s better to include free drug and free drug linker treatment control group in the cytotoxicity assay to prove the antiproliferative activity is due to NPs.
Ans.: As required, Table 2 of the revised version now reports the cytotoxicity data of free drugs and free drug linkers.
- The sizes of the NPs are all above 100 nm, some of them exceeds 300 nm, which means most of the NPs will be recognized by mononuclear phagocytes, preventing them from entering other tissues. Those NPs will be mostly in liver, spleen and blood. How is the cytotoxicity of these NPs in liver?
Ans.: Our group has monitored in the past [reference 4: Fumagalli et al. ChemPlusChem. 2015, 80, 1380-1383, doi:10.1002/cplu.201500156] the ability of hetero nanoparticles [mean diameter: (224±2) nm].to pass through the cell membrane in biological media, using direct stochastic optical reconstruction microscopy (dSTORM), providing the following results.
“Breast cancer MCF-7 cells were treated with the fluorescent hetero-nanoparticles for 20 min and imaged by multiple fluorescence microscopy methods. Three-dimensional confocal microscopy allowed the identification of the fluorescent nanoparticles in the cytoplasm of the cells, providing evidence that the nanoparticles are capable of permeating the cellular membrane. However, to provide an estimation of the size of the NPs, the resolution of conventional microscopy is not sufficient. We therefore applied super-resolution microscopy (dSTORM) to quantify the diameter of the bright particles detected inside the cells. The enlarged detail of the NP served to quantify its diameter: 156 nm. The diameter is in good agreement with our previous measurements by QELS [(224±2) nm]. In order to monitor the behaviour of the internalized NPs, we performed dStorm imaging at different incubation times. In this way we confirmed a progressive increase in the variability of the nanoparticle diameter—as indicated by the increase in the coefficient of variation—which might be an indication of their tendency to disassemble.”
- If the authors haven’t performed the in vivo study, is there any proof from literature indicating the efficacy and safety of such NPs?
Ans.: “For the most cytotoxic drugs 8 and 9, and for the corresponding NPs carrying the disulfide linker, 17bNP and 18bNP, the antiproliferative effect was also tested on human non tumorigenic Met-5A (mesothelium) cells. The obtained results evidence, as expected, a notable cytotoxicity induced by the drugs, with 0.0006 ± 0.0001 μM and 0.0032 ± 0.0013 μM GI50 values for 8 and 9, respectively; and confirm the decreased cell effect in the presence of 17bNP or 18bNP, i.e. GI50 0.11 ± 0.05 μM and 0.17 ± 0.04 μM respectively, in accordance with the results on human tumor cell lines” (lines 210-216).
Round 2
Reviewer 1 Report
The author provided all the details with respect comments, it can be accepted present form.
Reviewer 2 Report
The authors have improved the manuscript according to the suggestions of the reviewer and it is suitable for publication in the present form.
Reviewer 3 Report
Accept in present form